# A Million-Cow Validation of a Chromosome 14 Region Interacting with All Chromosomes for Fat Percentage in U.S. Holstein Cows

**DOI:** 10.3390/ijms25010674

**Published:** 2024-01-04

**Authors:** Dzianis Prakapenka, Zuoxiang Liang, Hafedh B. Zaabza, Paul M. VanRaden, Curtis P. Van Tassell, Yang Da

**Affiliations:** 1Department of Animal Science, University of Minnesota, Saint Paul, MN 55108, USA; 2Animal Genomics and Improvement Laboratory, USDA-ARS, Beltsville, MD 20705, USA

**Keywords:** GWAS, epistasis, fat percentage, SNP, Holstein, cow

## Abstract

A genome-wide association study (GWAS) of fat percentage (FPC) using 1,231,898 first lactation cows and 75,198 SNPs confirmed a previous result that a Chr14 region about 9.38 Mb in size (0.14–9.52 Mb) had significant inter-chromosome additive × additive (A×A) effects with all chromosomes and revealed many new such effects. This study divides this 9.38 Mb region into two sub-regions, Chr14a at 0.14–0.88 Mb (0.74 Mb in size) with 78% and Chr14b at 2.21–9.52 Mb (7.31 Mb in size) with 22% of the 2761 significant A×A effects. These two sub-regions were separated by a 1.3 Mb gap at 0.9–2.2 Mb without significant inter-chromosome A×A effects. The *PPP1R16A-FOXH1-CYHR1-TONSL* (PFCT) region of Chr14a (29 Kb in size) with four SNPs had the largest number of inter-chromosome A×A effects (1141 pairs) with all chromosomes, including the most significant inter-chromosome A×A effects. The *SLC4A4-GC-NPFFR2* (SGN) region of Chr06, known to have highly significant additive effects for some production, fertility and health traits, specifically interacted with the PFCT region and a Chr14a region with *CPSF1*, *ADCK5*, *SLC52A2*, *DGAT1*, *SMPD5* and *PARP10* (CASDSP) known to have highly significant additive effects for milk production traits. The most significant effects were between an SNP in SGN and four SNPs in PFCT. The CASDSP region mostly interacted with the SGN region. In the Chr14b region, the 2.28–2.42 Mb region (138.46 Kb in size) lacking coding genes had the largest cluster of A×A effects, interacting with seventeen chromosomes. The results from this study provide high-confidence evidence towards the understanding of the genetic mechanism of FPC in Holstein cows.

## 1. Introduction

The fat percentage (FPC) in Holstein cattle has the strongest genetic effects among dairy traits, and the gene of diacylglycerol O-acyltransferase 1 (*DGAT1*) of chromosome 14 (Chr14) has been widely confirmed to contain the most significant effects of FPC [1,2,3,4,5,6,7], including the effect of the *K232A* variant [3,4,5] and the effect of the single-nucleotide polymorphism (SNP) marker *rs109421300* (*ARS-BFGL-NGS-4939*) in *DGAT1* [1,6]. As an example showing how much more significant the FPC additive effects are than the additive effects of other dairy traits, the log_10_(1/p) value as a measure of statistical significance was 5150 for FPC, and it was 1320, 820, 374 and 371 for the protein percentage, milk yield, fat yield and protein yield, respectively, from a previous large-scale genome-wide association study (GWAS) using 60,671 SNPs and 294,079 Holstein cows (2019 GWAS) [1]. Although the exact reasons for the highly significant effect of *DGAT1* on FPC were not completely understood, the antagonism between milk and fat yields of *DGAT1* was a likely genetic mechanism [1,8,9], and the antagonism of the most significant SNP (*rs109421300*) was extreme, with the lowest milk yield and the highest fat yield among all SNPs on the cattle genome [1]. In addition, a large chromosome segment (6.79 Mb in size) containing *DGAT1* had highly significant effects on FPC. A conditional analysis showed that SNP effects in this region had strong linkage disequilibrium (LD) because the removal of the *rs109421300* genotypic values from the phenotypic values removed 82% of the significant effects in this region. A follow-up GWAS on epistasis effects showed that this Chr14 region was also involved in the genome-wide epistasis effects of FPC.

Using the same dataset of 294,079 Holstein cows and 76,109 SNPs from the 2019 GWAS for additive effects, genome-wide epistasis tests found that the Chr14 region with the most significant additive effects for FPC interacted with all chromosomes for FPC in the form of inter-chromosome additive × additive (A×A) effects [10]. This was a unique discovery among the five production traits with significant inter-chromosome epistasis effects, including FPC, protein percentage and milk, fat and protein yields. The significant epistasis effects in this 10 Mb region were divided into two sub-regions separated by a gap region without significant inter-chromosome A×A effects, a region including *DGAT1* upstream of the gap region and a region downstream of the gap region, which had over 40 coding genes but had no significant inter-chromosome A×A effects. The lack of significant inter-chromosome A×A effects in the gap region likely reflected the true status of this gap region: no A×A effects with other chromosomes. Given that both sides of this gap region had highly significant inter-chromosome A×A effects and the gap region had strong LD as shown by a previous conditional analysis [1], the lack of inter-chromosome A×A effects in the gap region was not due to the lack of LD with the two regions with significant inter-chromosome A×A effects on both sides of this gap region. Under the assumption of no A×A effects with other chromosomes, the statistical significance of this gap region provided a reference of the cut-off statistical significance to declare significant inter-chromosome A×A effects. The highest concentrations of the inter-chromosome A×A effects were in and around the *PPP1R16A* and *CYHR1* genes upstream of the gap region and were in a noncoding region immediately downstream of the gap region. Following the 2019 GWAS and the 2021 epistasis study, the Holstein cows that can be used for GWAS surpassed one million by the end of 2022. Such a large sample size should provide high statistical confidence to establish or deny all or any of the previous findings of the 10 Mb Chr14 region interacting with all chromosomes for FPC. For this purpose, we conducted the genome-wide validation of inter-chromosome A×A effects for FPC using over one million first lactation cows in this study.

## 2. Results and Discussion

The genome-wide tests of inter-chromosome A×A effects for FPC using 1,231,898 first lactation cows and 75,198 SNPs essentially confirmed the previous result that the 10 Mb Chr14 region interacted with all chromosomes for FPC, confirmed the structure of this 10 Mb region and revealed many new results. This study identified 2763 pairs of significant inter-chromosome A×A effects with log_10_(1/p) > 32 for FPC (Figure 1, Appendix A). Of these, 2761 pairs each involved the 0.14–9.52 Mb Chr14 region (9.38 Mb in size) with 107 SNPs and one of the remaining 29 chromosomes with 1050 SNPs, whereas only two pairs did not involve the Chr14 region, one pair between Chr03 and Chr05 and one pair between Chr05 and Chr20 (Figure 1a). Among the 2763 inter-chromosome A×A effects, the four most significant effects were between Chr06 and Chr14 (Figure 1b). The total number of SNPs in the 2763 SNP pairs was 1160. The circular plots of A×A effects between Chr14 and other chromosomes, as well as the Manhattan plots of the statistical significance of the A×A effects of all non-Chr14 chromosomes, are provided in Appendix A. Detailed test results, including statistical significance and estimated A×A effects and values for each of the 2763 SNP pairs, are provided in Appendix A.

### 2.1. Structure of the 9.38 Mb Chr14 Region Interacting with All Chromosomes for FPC

The significant inter-chromosome A×A effects in the 9.38 Mb Chr14 region (Figure 2a) were in two sub-regions that we named Chr14a and Chr14b, where Chr14a was the 0.14–0.88 Mb region (0.74 Mb in size) with 19 SNPs [11], and Chr14b was the 2.21–9.52 Mb region (7.31 Mb in size) with 88 SNPs [12]. These two sub-regions were separated by a 1.3 Mb gap at 0.9–2.2 Mb, without significant inter-chromosome A×A effects with log_10_(1/p) > 32 (Figure 2b) but with over 40 coding genes [13]. The Chr14a region had the largest number of significant inter-chromosome A×A effects, 2148 out of the 2761 pairs or 78%, interacting with all the remaining chromosomes, and the Chr14b region had 613 of the 2761 pairs or 22%, interacting with all the remaining chromosomes except Chr24 (Appendix A). Details of Chr14a and Chr14b are described below.

### 2.2. Inter-Chromosome A×A Effects of Chr14a

The Chr14a region was a gene-dense area with at least 46 coding genes [11], including the 15 genes shown in Figure 2c, which shows multiple locations with large clusters of inter-chromosome A×A effects. An SNP upstream of *LOC789384* (*rs136939758*) was the upstream boundary of Chr14a with 139 inter-chromosome A×A effects. An SNP in *PLEC* with one of the lowest log_10_(1/p) values (32.40) was the downstream boundary of Chr14a.

Two regions within Chr14a were particularly interesting: the *PPP1R16A-FOXH1-CYHR1-TONSL* (PFCT) region and the *CPSF1-ADCK5-SLC52A2-DGAT1-SMPD5-PARP10* (CASDSP) region. The PFCT region was only 29 Kb in size with four SNPs, but had the largest number of inter-chromosome A×A effects (1141 pairs) with all chromosomes, including the most significant inter-chromosome A×A effects (Figure 1b and Figure 2c, Table 1). The CASDSP region was 280 Kb in size, with six SNPs. Unlike PFCT, which interacted with all chromosomes, CASDPS mainly (42 out of 98 pairs) interacted with the *SLC4A4-GC-NPFFR2* (SGN) region of Chr06 (86.75–87.40 Mb). The epistasis effects between the SGN region of Chr06 and the PFCT and CASDSP regions of Chr14 are an important new discovery in this study.

### 2.3. Inter-Chromosome A×A Effects between Chr06 and Chr14a

The significant inter-chromosome A×A effects of Chr06 (314 effects) were distributed in the 6.11–114.38 Mb region (Figure 3a,b; Appendix A). However, Chr06 nearly exclusively interacted with Chr14a, with 312 of the 314 significant inter-chromosome A×A effects between Chr06 and Chr14a. Only two A×A effects involved Chr14b, between an SNP in the *ABCG2* gene of Chr06 and two SNPs in Chr14b (Figure 3a,b; Appendix A), noting that *ABCG2* had significant effect on milk yield [1]. Although the 312 pairs between Chr06 and Chr14a were distributed in the 6.11–114.38 Mb region, the A×A effects between Chr14a and the SGN region of chr06 were most interesting because the Chr14a and SGN regions were two of the most important regions affecting milk production, and the SGN region also had highly significant effects on somatic cell score, daughter pregnancy rate and cow conception rate [1,2,14,15,16,17].

The PFCT region had four SNPs each interacting with the same eleven SNPs in SGN, and the *CPSF1-ADCK5-SLC52A2-DGAT1* (CASD) region had three SNPs each interacting with the same nine SNPs in SGN (Figure 3c, Appendix A). The *SLC4A4* gene had ten significant SNPs and four of these ten SNPs in the 10.65 Kb tail region (86751807–86762457 bp), i.e., 2.5% of the gene interacted with both the PFCT and CASD regions of Chr14a, noting that the size of *SLC4A4* was 427.295 Kb. The *GC* gene had no significant inter-chromosome A×A effects for FPC. The *GC-NPFFR2* region had seven significant SNPs and two of these seven SNPs interacted with the PFCT region (Table 1). The top four most significant inter-chromosome A×A effects were between an SNP in *GC-NPFFR2* (*rs42766480*) of Chr06 and four SNPs in PFCT, *rs110984572* in *PPP1R16A-FOXH1* (#1), *rs137472016* in *CYHR1-TONSL* (#2), *rs137727465* in *CYHR1* (#3) and *rs10914637* in *PPP1R16A* (#4) (Table 1; Figure 3c,d). These results showed that *rs42766480* likely had a major role in the interactions with the PFCT region. However, this SNP was not ranked high (highest ranking #742) among the A×A effects between the SGN and CASD regions. The most significant inter-chromosome A×A effect between CASD and SGN was between *rs211309638* in *ADCK5-SLC52A2* and *rs110352004* in *GC-NPFFR2*, with a ranking of #37 among all effects. Other than this SNP, the effect rankings of SNPs in *CPSF1*, *DGAT1* and *SMPD5* were in the range of #153–#393 (Table 2), less significant than those of the PFCT region (ranking #1–#66, Table 1).

The most important feature of PFCT was that this small 29 Kb region of Chr14a interacted with all chromosomes. The most important feature of CASD was that this region mainly interacted with the SGN region of Chr06. The most important feature of the Chr06 inter-chromosome A×A effects was that the SGN region only interacted with Chr14a, including the PCFT and CASD regions. These results of Chr14a and the SGN region of Chr06 were particularly interesting because Chr14a and the SGN region of Chr06 were two of the most significant regions for milk production traits and the SGN region also was highly significant for somatic cell score and two fertility traits (daughter pregnancy rate and cow conception rate) [1,14].

### 2.4. Other Inter-Chromosome A×A Effects of Chr14a

The Chr14a region had other highly significant inter-chromosome A×A effects, in addition to those with the SGN region of Chr06, including those with Chr02, Chr05, Chr17, Chr29 and ChrX (Table 3, Figure 4). An SNP in *LOC789384* (*rs109208977*), an SNP between *ZNF250* and *ZNF16* (*rs110508680*) and an SNP in *ARHGAP39* each had a large cluster of inter-chromosome A×A effects, with 260, 195 and 231 inter-chromosome A×A effects, respectively (Figure 2c, Appendix A). Of the 20 highly significant effects not involving the SGN region of Chr06 (Table 3), 18 involved SNPs in the PCFT region, further showing a major role of the PCFT region in the inter-chromosome A×A effects for FPC.

### 2.5. Inter-Chromosome A×A Effects of Chr14b

The Chr14b region about 7.31 Mb in size (Figure 2b,d) [12] was nearly ten times as large as Chr14a (0.74 Mb in size) and had multiple locations with large clusters of inter-chromosome A×A effects. This region was divided into two sub-regions for convenience in describing the results: the 2.28–2.42 Mb region (138.46 Kb in size) as ‘Chr14b1’ and the remaining 7.17 Mb region as ‘Chr14b2’, with Chr14b1 accounting for 2% and Chr14b2 for 98% of Chr14b.

The Chr14b1 region had six significant SNPs in noncoding regions with two uncharacterized coding genes (*LOC112449593*, *LOC112449592*) and *TRNAC-GCA* [18]. These six SNPs had the largest number of inter-chromosome A×A effects (119 pairs) in Chr14b, involving seventeen chromosomes (chromosomes 2, 3, 4, 8, 12, 13, 16, 17, 18, 20, 21, 23, 25, 26, 28, 29, X) (Appendix A). The most significant inter-chromosome A×A effect of Chr14b (log_10_(1/p) = 64.37, #5 ranking) was between rs134537992 in Chr14b1 and rs42368654 about 96.6 Kb downstream of the *LMX1A* gene of Chr03 (Figure 5a, Table 4). This SNP of Chr03 interacted with 14 SNPs in Chr14b, two SNPs immediately downstream of this SNP interacted with an SNP in *KCNK9* in Chr14b2, and two other SNPs interacted with an SNP downstream of the *FAM135B* gene in Chr14b2. All the other Chr03 SNPs (96 total) interacted with Chr14a (Figure 5a, Appendix A). Although nearly the entire Chr03 interacted with Chr14a, the small region interacting with Chr14b1 had the most significant effect of Chr03. Other than the Chr03 region near *LMX1A*, the most significant effect of Chr14b1 was with Chr21 and Chr23 (Figure 5b,c; Table 4), noting that Chr23 almost interacted with Chr14a only, except for the inter-chromosome A×A effects with Chr14b1 (Figure 5c).

Based on the limited gene information in Chr14b1, two alternative hypotheses for the inter-chromosome A×A effects of the six SNPs in the Chr14b1 region could be made: (1) the noncoding sequences in the Chr14b1 region had biological functions in the form of interactions with other chromosomes for FPC, and (2) any or all *LOC112449593*, *LOC112449592* and *TRNAC-GCA* genes were responsible for the interactions between the six SNPs in the Chr14b1 region and the seventeen chromosomes due to LD with the significant SNPs. Hypothesis (1) should be the most likely reason for the interactions involving the six SNPs and implies major biological functions of the noncoding regions in the Chr14b1 region in the form of inter-chromosome A×A effects for FPC. Hypothesis (2) implies linked effects of the six SNPs through LD with any or all *LOC112449593*, *LOC112449592* and *TRNAC-GCA* genes. However, this hypothesis of linked effects was unlikely because the inter-chromosome A×A effects for FPC were unlikely affected in a significant way by LD with causal genes, as shown by the 1.3 Mb gap region of Chr14 (Figure 2b), which had at least 40 coding genes but no significant inter-chromosome A×A effects with log_10_(1/p) > 32.

The Chr14b2 region had significant inter-chromosome A×A effects mostly in or near four genes, *PTK2*, *TRAPPC9*, *KCNK9* and *FAM135B* (Figure 2d, Table 5). The *PTK2* gene interacted with eight chromosomes (chromosomes 2, 5, 10, 11, 16, 17, 19, 31), *TRAPPC9* with eight chromosomes (chromosomes 2, 5, 10, 11, 17, 20, 28, 31) and *KCNK9* with eleven chromosomes (chromosomes 2, 3, 4, 5, 8, 10, 19, 20, 21, 28, 31). This Chr14b2 region had many inter-chromosome A×A effects with ChrX (Figure 4d), which also interacted with Chr14a and Chr14b1. Chr10 almost exclusively interacted with Chr14b2 (Figure 5d). Of the 55 inter-chromosome A×A effects of Chr14b, only six effects were between Chr14a and four SNPs of Chr10, including those between an SNP in *GNG2* of Chr10 and three SNPs in *DGAT1*, *PARP10* and *PLEC* of Chr14a, and between an SNP in *LOC789384* of Chr14a and three SNPs of Chr10 (Appendix A). The remaining 49 inter-chromosome A×A effects of Chr14b2, except one, were in or near *PTK2*, *TRAPPC9*, *KCNK9* and *GPR20* (Appendix A).

### 2.6. Inter-Chromosome A×A Effects of Chr20 and Chr05 Interacting with Chr14

Chr20 and Chr05, along with Chr14 and Chr06, also had highly significant additive effects for milk production traits. Therefore, it was of interest to determine whether the Chr20 and Chr05 regions affecting milk production traits also interacted with the Chr14 region for FPC.

The inter-chromosome A×A effects of Chr20 covered a large distance of 63.52 Mb (6.58–70.10 Mb). Chr14a interacted with the 6.58–28.8 Mb region (mostly the 20–28 Mb region) and Chr14b with the 30.61–42.14 Mb region, whereas both Chr14a and Chr14b interacted with the remaining regions of Chr20 (Figure 6a). The most significant inter-chromosome A×A effect of Chr20 (log_10_(1/p) = 57.12) was that between *rs136653182* about 332 Kb downstream of the *ITGA1* gene of Chr20 and *rs109208977* in *LOC789384* of Chr14a (Figure 6a, Table 6). The 20–28 Mb region of Chr20 interacting with Chr14a was near the location with the most significant effects of Chr20 at 31–33 Mb. In contrast, the 30.61–42.14 Mb region of Chr20 interacting with Chr14b had the most significant effects for milk yield among Chr20 SNPs. In particular, the *NNT* gene had highly significant effects for milk yield and had an SNP interacting with two SNPs in the Chr14b1 region that had the largest cluster of inter-chromosome A×A effects of Chr14b (Figure 4b, Table 3). The most significant A×A effect in the 30.61–42.14 Mb region of Chr20 was that between *rs133536911* about 10.59 Kb downstream of the *FGF10* gene of Chr20 and *rs134537992* in Chr14b1, and *rs133536911* also interacted with four other SNPs of Chr14b (Appendix A). These results showed that the inter-chromosome A×A effects between Chr20 and the Chr14 region involved the Chr20 region with highly significant effects for milk yield.

The inter-chromosome A×A effects of Chr05 were mostly in the 0.5–10.8 Mb region interacting with Chr14a (Figure 4c, Table 3). The most significant SNP of chr05 was an intergenic SNP (*rs109208465*) about 71.13 Kb upstream of the *BBS10* gene that interacted with six SNPs in PFCT (log_10_(1/p) = 59.25–62.89) and four SNPs in CASD and *SMPD5* (log_10_(1/p) = 32.40–34.56) of Chr14a. However, the 0.5–10.8 Mb Chr05 region did not have highly significant effects for milk and fat yields or FPC [1]. The *MGST1-SLC15A5* region (93.51–93.63 Mb) of Chr05 had highly significant additive effects on fat yield and FPC but this region did not interact with Chr14a or Chr14b for FPC. The SNP closest to the *MGST1-SLC15A5* region was *rs134855280* at 92.59 Mb, which had a significant inter-chromosome A×A effect with an SNP in *LOC789384* of Chr14a. It was interesting that an SNP in *EPS8* about 701 Kb downstream of the *MGST1-SLC15A5* region had inter-chromosome A×A effects with an SNP in Chr03 and an SNP in Chr20, and these two inter-chromosome A×A effects were the only ones not involving Chr14a or Chr14b (Appendix A), noting that *EPS8* had highly significant additive effects for FPC. The 23–44 Mb region had significant SNP additive effects for milk and fat yields, and this large region interacted with both Chr14a and Chr14b for FPC (Appendix A).

### 2.7. Patterns of A×A Epistasis Effects

The A×A values of the four allelic combinations (*AB*, *Ab*, *aB*, *ab*) of each pair of loci typically had large absolute values for the most positive and negative allelic combinations. Let AC1, AC2, AC3 and AC4 represent the four allelic combinations from the most positive combination to most negative combination and let aa1-aa4 represent the A×A values of AC1-AC4, where ‘AC’ stands for ‘allelic combination’. Then, AC1 and AC4 had the largest absolute A×A values, whereas AC2 and AC3 had considerably smaller absolute A×A values than those of AC1 and AC4 (Appendix A, Table 7 and Table 8). Consequently, the size of the A×A effect of two loci as a contrast of aa1-aa4 (Equations (1) and (2) in Materials and Methods) was mostly determined by the A×A values of AC1 and AC4. Therefore, the discussion of A×A patterns focused on the A×A values of AC1 and AC4, which had two patterns: (1) the two A×A values involved the same chr14 allele and two non-Chr14 alleles such as the 1_1 allelic combination for AC1 and 1_2 allelic combination for AC4, and (2) the A×A values involved two chr14 alleles and the same non-Chr14 allele, such as 1_1 for AC1 and 2_1 for AC4. Most Chr14a A×A values (1554 out of 2148, or 72%) had pattern (1), whereas most Chr14b A×A values (346 out of 613, or 56%) had pattern (2). It was interesting that no AC1 and AC4 of any SNP pair involved completely different alleles, such as 1_1 for AC1 and 2_2 for AC4, or 1_2 for AC1 and 2_1 for AC4.

Table 7 shows examples of the Chr14a A×A values where the AC1 and AC4 of each SNP pair had the same Chr14a allele and two different non-Chr14 alleles. SNP *rs109421300* of *DGAT1* should be a highly recognizable SNP because this SNP had the most significant effects for all five production traits: milk, fat and protein yields and fat and protein percentages. Allele 1 of *rs109421300* had an extreme antagonism between fat yield and milk and protein yields, with the most positive effect for fat yield and FPC and most negative effects for milk and protein yields [1]. In this study, allele 1 of *rs109421300* was the common Chr14a allele of AC1 and AC4 for all nine pairs of A×A values, and each of the nine SNPs in the SGN region of Chr06 had both alleles in AC1 and AC4. The AC1 and AC4 for seven of the nine SNP pairs had similar absolute values, indicating that these AC1 and AC4 were approximately symmetric. The combination of allele 1 of *rs109421300* with a Chr06 allele was positive (aa1, Table 7), whereas the combination of allele 1 of *rs109421300* with the alternative Chr06 allele of each Chr06 SNP was negative (aa4, Table 7). The other A×A values of Chr14a had similar patterns. The results of Table 7 indicate that one allele of a Chr14a SNP interacted with both alleles of a non-Chr14 SNP for most of the significant SNP pairs involving Chr14a.

Table 8 shows examples of the Chr14b A×A values where the AC1 and AC4 of each SNP pair had the same non-Chr14 allele and two different Chr14b alleles. Allele 1 of SNP *rs42368654* downstream of *LMX1A* of Chr03 was the common allele of AC1 and AC4 with five SNPs of Chr14b1.The size of AC1 was 2–5 times as large as that of AC4 for the five A×A values, indicating that the interaction between allele 1 of *rs42368654* of Chr03 and one Chr14b1 was the main contributor of the five A×A values. Of the twenty SNP pairs in Table 8, the size of AC1 was larger than that of AC4 for eighteen pairs, AC4 was larger than AC1 for two pairs, and AC1 and AC2 had approximately the same sizes for two pairs.

## 3. Materials and Methods

### 3.1. Holstein Population and SNP Data

The Holstein population in this study had 1,231,898 first lactation cows with phenotypic observations of fat percentage (FPC) and genotypes of 78,964 original and imputed SNPs. The SNP genotypes were from 32 SNP chips with various densities and were imputed to 78,964 SNPs via the FindHap algorithm [19] as a routine procedure for genomic evaluation by the Council on Dairy Cattle Breeding (CDCB) [20]. The phenotypic values used in the GWAS analysis were the phenotypic residuals after removing fixed non-genetic effects available from the December 2022 U.S. Holstein genomic evaluation by the CDCB. Basic statistics of the cows and phenotypic data of FPC are given in Appendix A. With the requirement of a 0.05 minor allele frequency, the number of SNPs for the GWAS analysis was 75,198 SNPs. A strict criterion of log_10_(1/p) > 32 was used to declare the statistical significance of any inter-chromosome epistasis effect. This requirement ensured that any significant effect had better statistical significance than that of the highest statistical significance of log_10_(1/p) = 32 shown in Figure 2a,b. The log_10_(1/p) > 32 requirement was stricter than the requirement of log_10_(1/p) > 12 for the Bonferroni correction with 0.05 genome-wide false positives. The SNP and gene positions were those from the ARS-UCD1.3 cattle genome assembly [21]. Genes containing or in the proximity of highly significant effects were identified as candidate genes affecting FPC.

### 3.2. GWAS Analysis

The A×A value of each of the four allelic combinations (*AB*, *Ab*, *aB*, *ab*) of two loci was calculated as the deviation of the mean of the allelic combination from the population mean and the additive values of the two alleles in the allelic combination [22,23]:(1)$${\left(\mathrm{aa}\right)}_{\mathrm{ik}}=\mu_{\mathrm{ik}}-\mu -a_{i}-a_{k}$$
where ${\left(\mathrm{aa}\right)}_{\mathrm{ik}}$ = A×A value of allelic combination of the $i^{\mathrm{th}}$ allele of locus 1 (*A* or *a* allele) and the $k^{\mathrm{th}}$ allele of locus 2 (*B* or *b* allele), $\mu_{\mathrm{ik}}$ = the mean of the genotypic values with allelic combination of the $i^{\mathrm{th}}$ allele of locus 1 (*A* or *a* allele) and the $k^{\mathrm{th}}$ allele of locus 2 (*B* or *b* allele), μ = the population mean of genotypic values, $a_{i}=\mu_{i}-\mu$ (i = *A* or *a*) = additive value of $i^{\mathrm{th}}$ allele of locus 1 (*A* or *a* allele), $a_{k}=\mu_{k}-\mu$ (k = *B* or *b*) = additive value of the $k^{\mathrm{th}}$ allele of locus 2 (*B* or *b* allele), $\mu_{i}$ = the mean of genotypic values with the $i^{\mathrm{th}}$ allele of locus 1 (*A* or *a* allele), and $\mu_{k}$ = the mean of genotypic values with the $k^{\mathrm{th}}$ allele of locus 2 (*B* or *b* allele).

The A×A effect of two loci was calculated as a contrast of the four A×A values and this contrast was further expressed as the A×A contrast of the nine genotypic values for epistasis testing [24]: (2)$$\begin{array}{l} \alpha \alpha =[{\left(\mathrm{aa}\right)}_{AB}-{\left(\mathrm{aa}\right)}_{Ab}]-[{\left(\mathrm{aa}\right)}_{aB}-{\left(\mathrm{aa}\right)}_{ab}]\\ \;\;\;\;=[{\left(\mathrm{aa}\right)}_{AB}-{\left(\mathrm{aa}\right)}_{aB}]-[{\left(\mathrm{aa}\right)}_{Ab}-{\left(\mathrm{aa}\right)}_{ab}]\\ \;\;\;\;=(\mu_{AB}-\mu_{Ab})-(\mu_{aB}-\mu_{ab})\\ \;\;\;\;=(\mu_{AB}-\mu_{aB})-(\mu_{Ab}-\mu_{ab})\\ \;\;\;\;=\;L_{a\times a}=s_{a\times a}^{}g \end{array}$$
where αα = A×A effect of the two loci as a contrast of the four A×A values of the four allelic combinations of the two loci; ${(\mathrm{aa})}_{AB}$, ${\left(\mathrm{aa}\right)}_{Ab}$, ${\left(\mathrm{aa}\right)}_{aB}$ and ${\left(\mathrm{aa}\right)}_{ab}$ are the four A×A values of the four allelic combinations of AB, Ab, aB and ab, respectively, defined by Equation (1); $\mu_{AB}$= the mean of genotypic values with the AB allelic combination, $\mu_{Ab}$= the mean of genotypic values with the Ab allelic combination, $\mu_{aB}$ = the mean of genotypic values with the aB allelic combination, $\mu_{ab}$ = the mean of genotypic values with the ab allelic combination; g = column vector of the nine SNP genotypic values of the two loci: $g_{AABB}$, $g_{AABb}$, $g_{AAbb}$, $g_{AaBB}$, $g_{AaBb}$, $g_{Aabb}$, $g_{aaBB}$, $g_{aaBb}$, $g_{aabb}$; $s_{a\times a}^{}$ = row vector of the A×A contrast coefficients of nine SNP genotypic values; and $\;L_{a\times a}$ = A×A effect of the two loci as a contrast of the nine SNP genotypic values. In the absence of allelic interactions between the two loci, the A×A effect of Equation (2) is expected to be null because each A×A value is expected to be null. In the presence of an allele × allele interaction between the two loci, the $\left[{(\mathrm{aa})}_{AB}-{\left(\mathrm{aa}\right)}_{Ab}]-[{\left(\mathrm{aa}\right)}_{aB}-{\left(\mathrm{aa}\right)}_{ab}\right]$ definition of the A×A effect indicates that the allelic difference of locus 2 changes in the presence of the two different alleles of locus 1, whereas the $\left[{\left(\mathrm{aa}\right)}_{AB}-{\left(\mathrm{aa}\right)}_{aB}]-[{\left(\mathrm{aa}\right)}_{Ab}-{\left(\mathrm{aa}\right)}_{ab}\right]$ definition of A×A effect indicates that the allelic difference of locus 1 changes in the presence of the two different alleles of locus 2. Therefore, a significant A×A effect expressed as $L_{a\times a}=s_{a\times a}^{}g$ indicates the presence of an allele × allele interaction between the two loci due to the equivalence between this expression and any of the other four expressions in Equation (2).

The GWAS analysis of A×A effects used an approximate generalized least squares (AGLS) method. The AGLS method combines the least squares (LS) tests implemented by EPISNP1mpi [25,26] with the estimated breeding values from routine genetic evaluation using the entire U.S. Holstein population. The statistical model was
(3)$$y=\mu I+X_{g}g+Za+e=Xb+Za+e$$
where **y** = column vector of phenotypic deviation after removing fixed nongenetic effects such as heard-year-season (termed as ‘yield deviation’ for any trait) using a standard procedure for the CDCB/USDA genetic and genomic evaluation; µ = common mean; **I** = identity matrix; **g** = column vector of genotypic values; $X_{g}$ = model matrix of **g**; b=(μ, g′)′, $X=(I,\;X_{g})$; **a** = column vector of additive polygenic values; **Z** = model matrix of **a**; and **e** = column vector of random residuals. The first and second moments of Equation (3) are E(y)=Xb and var(y)=V=ZGZ′+R = $\sigma_{a}^{2}ZAZ\prime +\sigma_{e}^{2}I$, where $\sigma_{a}^{2}$ = additive variance, **A** = additive relationship matrix and $\sigma_{e}^{2}$ = residual variance. The problem of estimating the **b** vector that includes SNP genotypic values in Equation (3) is the requirement of inverting the **V** if the generalized least squares (GLS) method is used, or inverting the **A** matrix and the coefficient matrix of the mixed model equations (MME) if the MME method is used [27]. However, both **V** and MME could not be inverted for our sample size. To avoid inverting these large matrices, the GWAS used the method of approximate GLS (AGLS), which replaces the polygenic additive values (**a**) with the best linear unbiased prediction based on pedigree relationships [1]. The AGLS method is based on the following results:(4)$$\hat{b}={\left(X\prime V^{-1}X\right)}^{-}X\prime V^{-1}y$$
(5)$$\begin{array}{l} \hat{b}={\left(X\prime R^{-1}X\right)}^{-}(X\prime R^{-1}y-X\prime R^{-1}Z\hat{a})\\ \;\;\;\;={\left(X\prime X\right)}^{-}X\prime (y-Z\hat{a})={\left(X\prime X\right)}^{-}X\prime y_{*} \end{array}$$
where $y_{*}=y-Z\hat{a}$ and $\hat{a}$ is the best linear unbiased prediction (BLUP) of **a**. Equation (4) is the GLS solution, and Equation (5) is the MME solution of **b**. These two equations yield identical results, and $\hat{b}$ from either equation is termed the best linear unbiased estimator (BLUE) [27]. If $\hat{a}$ is known, the LS version of BLUE given by Equation (5) is computationally efficient relative to the GLS of Equation (4), requiring the **V** inverse, or the joint MME solutions of $\hat{b}$ and $\hat{a}$, requiring the inverse of the coefficient matrix of the MME. The AGLS method uses two approximations. The first approximation is to use $\tilde{a}$ from routine genetic evaluation as an approximation of $\hat{a}$ in Equation (5):(6)$$\hat{b}={\left(X\prime X\right)}^{-}X\prime (y-Z\tilde{a})={\left(X\prime X\right)}^{-}X\prime y^{*}$$
where $y^{*}=y-Z\tilde{a}$, and $\tilde{a}$ is the column vector of 2(PTA), with PTA being the predicted transmission ability from the routine genetic evaluation. Equation (6) achieves the benefit of sample stratification correction from mixed models using pedigree relationships without the computing difficulty of inverting **V** or **A**. The second approximation of the AGLS approach is the *t*-test using the LS rather than the GLS formula of the t-statistic, to avoid using the **V** inverse in the GLS formula. The significance tests for A×A SNP effects used the *t*-tests of the A×A contrast of the estimated two-locus SNP genotypic values [24,25]: (7)$$t_{a\times a}=\frac{{|L}_{a\times a}|}{\sqrt{{\mathrm{var}(L}_{a\times a})}}=\frac{\left|s_{a\times a}^{}\hat{g}\right|}{v\sqrt{s_{a\times a}^{}{\left(X_{}^{\prime }X_{}\right)}_{\mathrm{gg}}^{-}s_{a\times a}^{\prime }}}$$
where $L_{a\times a}$ = A×A contrast of the nine genotypic values defined by Equation (1); $\sqrt{{\mathrm{var}(L}_{a\times a})}$ = standard deviation of $L_{a\times a}$; $s_{a\times a}^{}$ = row vector of A×A contrast coefficients; $v^{2}=\left(y-X\hat{b}{)}^{\prime }(y-X\hat{b}\right)/(n-k)$= estimated residual variance; $\hat{g}$ = column vector of the AGLS estimates of the nine SNP genotypic values of the two loci; and ${\left(X_{}^{\prime }X_{}\right)}_{\mathrm{gg}}^{-}$ = submatrix of ${\left(X_{}^{\prime }X\right)}_{}^{-}$ corresponding to $\hat{g}$.

## 4. Conclusions

This GWAS using over 1.2 million Holstein cows confirmed that a Chr14 region about 9.38 Mb region in size had significant inter-chromosome additive × additive (A×A) effects with all chromosomes for FPC in two sub-regions separated by a gap region without significant inter-chromosome A×A effects. Inside this 9.38 Mb region, a 0.75 Mb region known to have highly significant additive effects of FPC had most of the inter-chromosome A×A effects, including those with a Chr06 region that was known to have highly significant additive effects for some production, reproduction and health traits. This GWAS using an unprecedentedly large sample provides high-confidence evidence that FPC is affected by genome-wide allele × allele interactions centered in the 9.38 Mb Chr14 region.

## Figures and Tables

**Figure 1 ijms-25-00674-f001:**
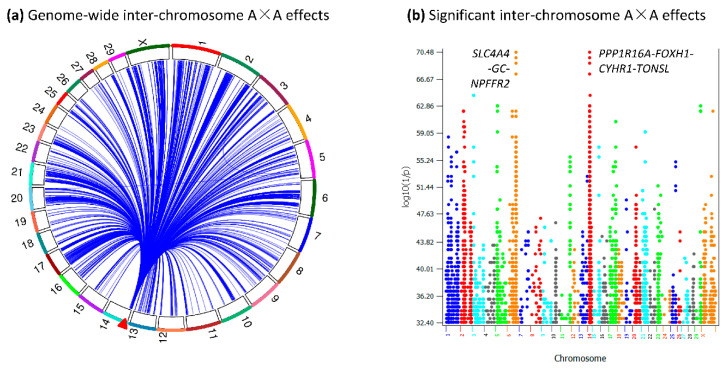
Significant inter-chromosome A×A effects of FPC (log_10_(1/p) > 32). (**a**) The 0.14–9.52 Mb region of Chr14 (marked by the red arrow) had significant inter-chromosome A×A effects with all chromosomes. (**b**) Significant inter-chromosome A×A effects of each chromosome.

**Figure 2 ijms-25-00674-f002:**
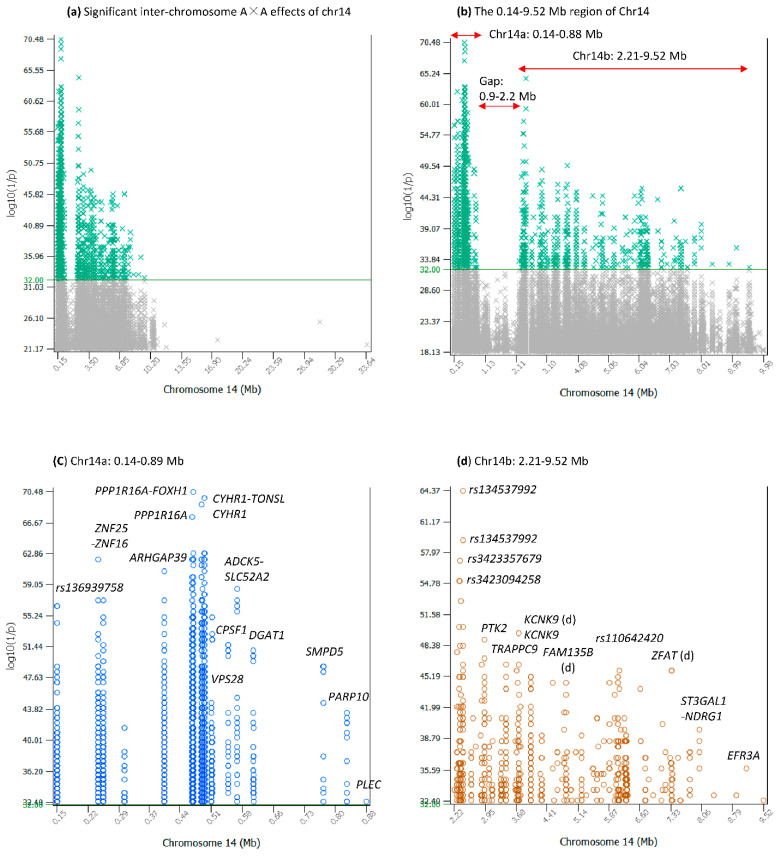
Inter-chromosome A×A effects of Chr14. (**a**) Significant inter-chromosome A×A effects of Chr14 among the top 50,000 pairs of inter-chromosome A×A effects. Effects with log_10_(1/p) > 32 were considered statistically significant. (**b**) Inter-chromosome A×A effects of the 0.14–9.52 Mb region with two sub-regions of Chr14a and Chr14b that were separated by a 1.3 Mb gap region. Effects with log_10_(1/p) > 32 were considered statistically significant. (**c**) Inter-chromosome A×A effects of Chr14a with log_10_(1/p) > 32. (**d**) Inter-chromosome A×A effects of Chr14b with log_10_(1/p) > 32.

**Figure 3 ijms-25-00674-f003:**
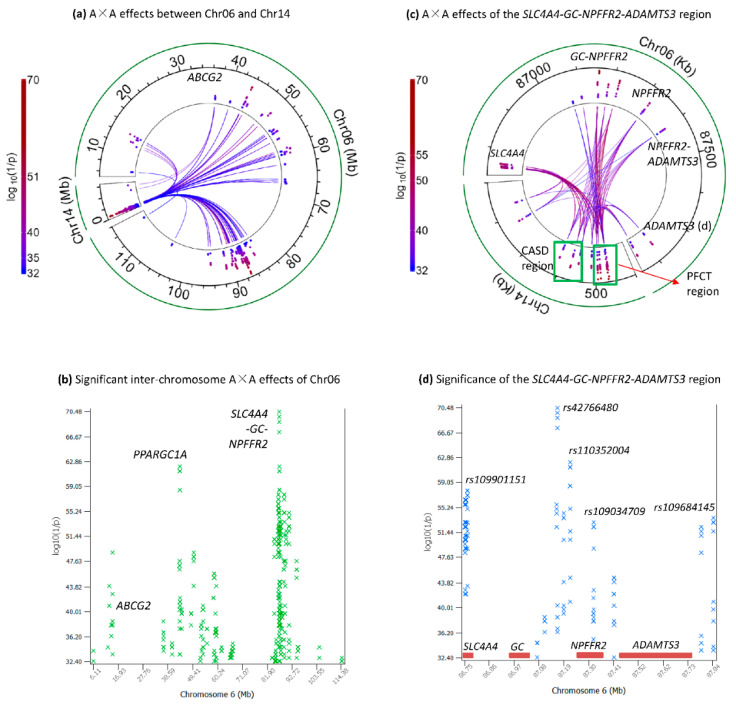
Significant inter-chromosome A×A effects of FPC between Chr14a and Chr06 with log_10_(1/p) > 32. (**a**) Inter-chromosome A×A effects of Chr06. (**b**) Inter-chromosome A×A effects between Chr14a and the SGN region Chr06. (**c**) Manhattan plot of statistical significance of inter-chromosome A×A effects of Chr06. (**d**) Manhattan plot of statistical significance of inter-chromosome A×A effects between Chr14a and the SGN region Chr06.

**Figure 4 ijms-25-00674-f004:**
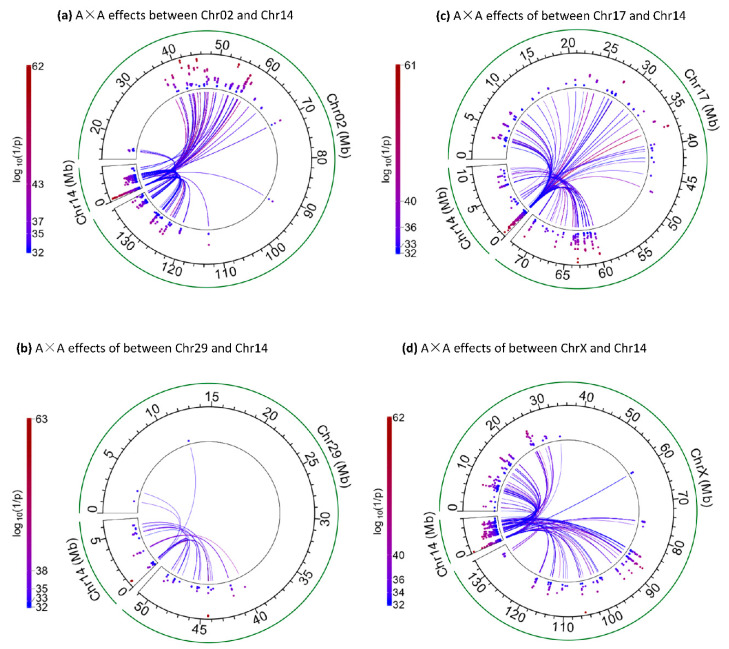
Examples of significant inter-chromosome A×A effects of Chr14 for FPC with log_10_(1/p) > 32. (**a**) Inter-chromosome A×A effects of Chr02. (**b**) Inter-chromosome A×A effects of Chr29. (**c**) Inter-chromosome A×A effects of Chr17. (**d**) Inter-chromosome A×A effects of ChrX.

**Figure 5 ijms-25-00674-f005:**
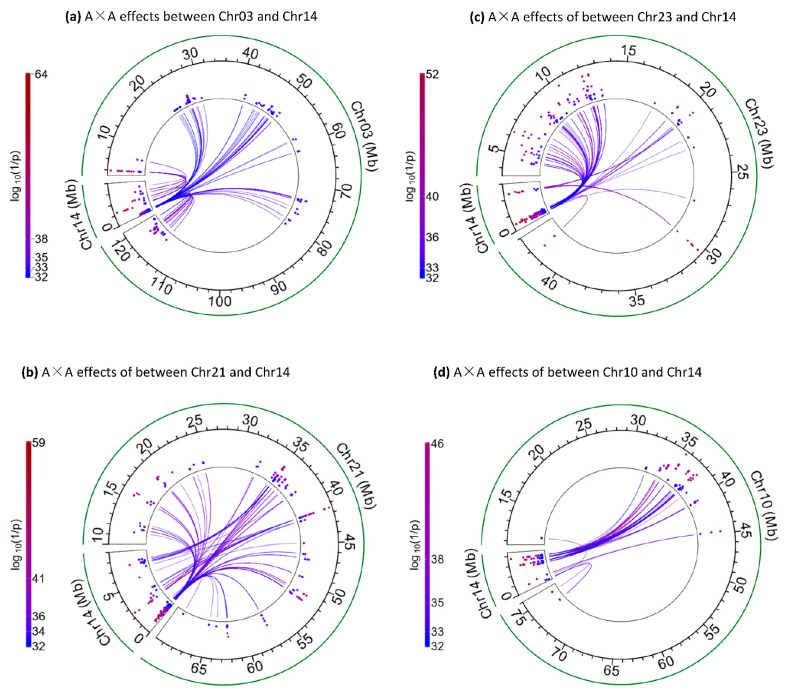
Inter-chromosome A×A effects of Chr03 and Chr10 for FPC. (**a**) Inter-chromosome A×A effects between Chr03 and Chr14. (**b**) Inter-chromosome A×A effects between Chr21 and Chr14. (**c**) Inter-chromosome A×A effects between Chr23 and Chr14. (**d**) Inter-chromosome A×A effects between Chr10 and Chr14.

**Figure 6 ijms-25-00674-f006:**
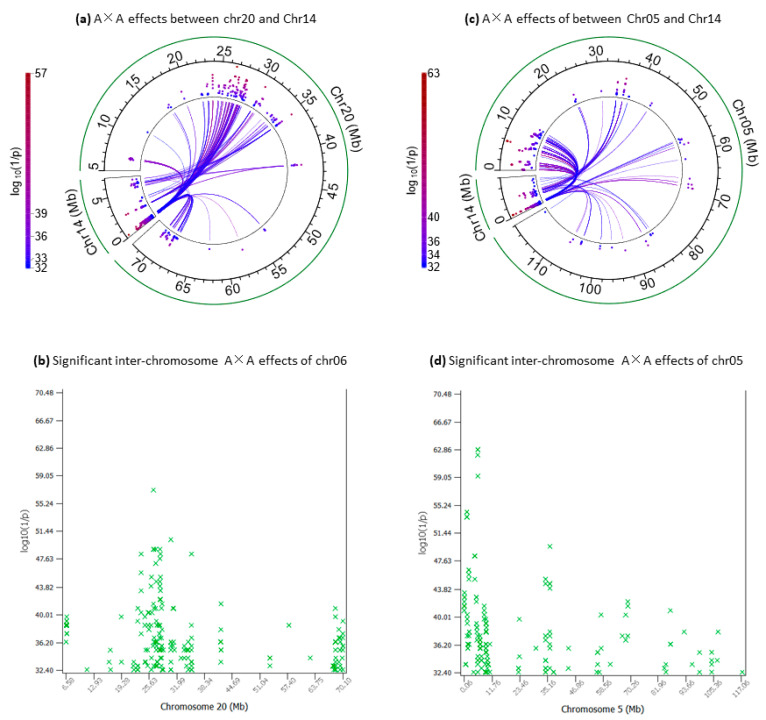
Inter-chromosome A×A effects of Chr20 and Chr05 for FPC. (**a**) Inter-chromosome A×A effects between Chr20 and Chr14. (**b**) Manhattan plot of statistical significance of inter-chromosome A×A effects of Chr20. (**c**) Inter-chromosome A×A effects between Chr05 and Chr14. (**d**) Manhattan plot of statistical significance of inter-chromosome A×A effects of Chr05.

**Table 1 ijms-25-00674-t001:** Inter-chromosome A×A effects between four SNPs in the PFCT region of Chr14a and six SNPs in *SLC4A4- NPFFR2* of Chr06.

SNP-1	Chr-1	Pos-1	Gene-1	SNP-2	Chr-2	Pos-2	Gene-2	Effect	log10(1/p)	Rank
*rs109146371*	14	465742	*PPP1R16A*	*rs42766480*	6	87156735	*GC-NPFFR2*	0.0062	67.39	4
*rs109146371*	14	465742	*PPP1R16A*	*rs110352004*	6	87213962	*GC-NPFFR2*	−0.0058	58.53	35
*rs110984572*	14	468124	*PPP1R16A-FOXH1*	*rs137302420*	6	86751807	*SLC4A4*	−0.0058	55.72	59
*rs110984572*	14	468124	*PPP1R16A-FOXH1*	*rs109512265*	6	86753255	*SLC4A4*	0.0058	56.42	53
*rs110984572*	14	468124	*PPP1R16A-FOXH1*	*rs110953922*	6	86755896	*SLC4A4*	0.0058	56.42	54
*rs110984572*	14	468124	*PPP1R16A-FOXH1*	*rs109901151*	6	86762457	*SLC4A4*	0.0058	57.82	38
*rs110984572*	14	468124	*PPP1R16A-FOXH1*	*rs42766480*	6	87156735	*GC-NPFFR2*	−0.0064	70.48	1
*rs110984572*	14	468124	*PPP1R16A-FOXH1*	*rs110352004*	6	87213962	*GC-NPFFR2*	0.0060	62.15	12
*rs137727465*	14	487527	*CYHR1*	*rs137302420*	6	86751807	*SLC4A4*	0.0057	55.72	61
*rs137727465*	14	487527	*CYHR1*	*rs109901151*	6	86762457	*SLC4A4*	−0.0057	57.12	46
*rs137727465*	14	487527	*CYHR1*	*rs42766480*	6	87156735	*GC-NPFFR2*	0.0063	68.92	3
*rs137727465*	14	487527	*CYHR1*	*rs110352004*	6	87213962	*GC-NPFFR2*	−0.0059	61.42	21
*rs137472016*	14	494621	*CYHR1-TONSL*	*rs137302420*	6	86751807	*SLC4A4*	−0.0058	55.72	66
*rs137472016*	14	494621	*CYHR1-TONSL*	*rs109512265*	6	86753255	*SLC4A4*	0.0058	56.42	55
*rs137472016*	14	494621	*CYHR1-TONSL*	*rs110953922*	6	86755896	*SLC4A4*	0.0058	56.42	56
*rs137472016*	14	494621	*CYHR1-TONSL*	*rs109901151*	6	86762457	*SLC4A4*	0.0058	57.82	40
*rs137472016*	14	494621	*CYHR1-TONSL*	*rs42766480*	6	87156735	*GC-NPFFR2*	−0.0064	69.70	2
*rs137472016*	14	494621	*CYHR1-TONSL*	*rs110352004*	6	87213962	*GC-NPFFR2*	0.0060	61.42	22

Pos is the chromosome position of the SNP.

**Table 2 ijms-25-00674-t002:** Inter-chromosome A×A effects between four SNPs in CASDS region of Chr14a and the SGN region of Chr06.

SNP-1	Chr-1	Pos-1	Gene-1	SNP-2	Chr-2	Pos-2	Gene-2	Effect	log10(1/p)	Rank
*rs134432442*	14	550784	*CPSF1*	*rs137302420*	6	86751807	*SLC4A4*	−0.0056	50.30	178
*rs134432442*	14	550784	*CPSF1*	*rs109512265*	6	86753255	*SLC4A4*	0.0056	50.96	164
*rs134432442*	14	550784	*CPSF1*	*rs110953922*	6	86755896	*SLC4A4*	0.0056	50.96	165
*rs134432442*	14	550784	*CPSF1*	*rs109901151*	6	86762457	*SLC4A4*	0.0055	51.63	153
*rs134432442*	14	550784	*CPSF1*	*rs110352004*	6	87213962	*GC-NPFFR2*	0.0055	51.63	154
*rs211309638*	14	572120	*ADCK5-SLC52A2*	*rs137302420*	6	86751807	*SLC4A4*	−0.0059	55.72	68
*rs211309638*	14	572120	*ADCK5-SLC52A2*	*rs109512265*	6	86753255	*SLC4A4*	0.0059	56.42	57
*rs211309638*	14	572120	*ADCK5-SLC52A2*	*rs110953922*	6	86755896	*SLC4A4*	0.0059	56.42	58
*rs211309638*	14	572120	*ADCK5-SLC52A2*	*rs109901151*	6	86762457	*SLC4A4*	0.0059	57.12	49
*rs211309638*	14	572120	*ADCK5-SLC52A2*	*rs110352004*	6	87213962	*GC-NPFFR2*	0.0060	58.53	37
*rs109421300*	14	609870	*DGAT1*	*rs137302420*	6	86751807	*SLC4A4*	0.0055	49.64	191
*rs109421300*	14	609870	*DGAT1*	*rs109512265*	6	86753255	*SLC4A4*	−0.0056	50.30	179
*rs109421300*	14	609870	*DGAT1*	*rs110953922*	6	86755896	*SLC4A4*	−0.0055	50.30	180
*rs109421300*	14	609870	*DGAT1*	*rs109901151*	6	86762457	*SLC4A4*	−0.0055	50.96	166
*rs109421300*	14	609870	*DGAT1*	*rs110352004*	6	87213962	*GC-NPFFR2*	−0.0055	50.30	181
*rs135549651*	14	775260	*SMPD5*	*rs137302420*	6	86751807	*SLC4A4*	−0.0054	48.33	240
*rs135549651*	14	775260	*SMPD5*	*rs109512265*	6	86753255	*SLC4A4*	0.0054	48.98	211
*rs135549651*	14	775260	*SMPD5*	*rs110953922*	6	86755896	*SLC4A4*	0.0054	48.98	212
*rs135549651*	14	775260	*SMPD5*	*rs109901151*	6	86762457	*SLC4A4*	0.0053	48.98	213
*rs135549651*	14	775260	*SMPD5*	*rs110352004*	6	87213962	*GC-NPFFR2*	0.0051	44.52	393

Pos is the chromosome position of the SNP.

**Table 3 ijms-25-00674-t003:** Top inter-chromosome A×A effects of the Chr14 excluding those with the SGN region of Chr06.

SNP-1	Chr-1	Pos-1	Gene-1	SNP-2	Chr-2	Pos-2	Gene-2	Effect	log10(1/p)	Rank
*rs110984572*	14	468124	*PPP1R16A-FOXH1*	*rs109208465*	5	5756462	*PHLDA1-BBS10*	0.006	62.89	6
*rs137472016*	14	494621	*CYHR1-TONSL*	*rs109208465*	5	5756462	*PHLDA1-BBS10*	0.006	62.89	7
*rs137472016*	14	494621	*CYHR1-TONSL*	*rs109210391*	29	44322319	*KLC2*	0.013	62.89	8
*rs110984572*	14	468124	*PPP1R16A-FOXH1*	*rs43304498*	2	41451308	*KCNJ3-GALNT13*	0.009	62.15	9
*rs137727465*	14	487527	*CYHR1*	*rs43304498*	2	41451308	*KCNJ3-GALNT13*	−0.009	62.15	10
*rs137472016*	14	494621	*CYHR1-TONSL*	*rs43304498*	2	41451308	*KCNJ3-GALNT13*	0.009	62.15	11
*rs137727465*	14	487527	*CYHR1*	*rs109208465*	5	5756462	*PHLDA1-BBS10*	−0.006	62.15	12
*rs137472016*	14	494621	*CYHR1-TONSL*	*rs41596003*	6	43758146	*PPARGC1A*	0.006	62.15	13
*rs109146371*	14	465742	*PPP1R16A*	*rs109210391*	29	44322319	*KLC2*	−0.013	62.15	15
*rs110984572*	14	468124	*PPP1R16A-FOXH1*	*rs109210391*	29	44322319	*KLC2*	0.013	62.15	16
*rs137727465*	14	487527	*CYHR1*	*rs109210391*	29	44322319	*KLC2*	−0.013	62.15	17
*rs109208977*	14	243959	*LOC789384*	*rs134212233*	31	105004567	*ENSBTAG00000045867*	0.006	62.15	18
*rs110984572*	14	468124	*PPP1R16A-FOXH1*	*rs41596003*	6	43758146	*PPARGC1A*	0.006	61.42	19
*rs137727465*	14	487527	*CYHR1*	*rs41596003*	6	43758146	*PPARGC1A*	−0.006	61.42	20
*rs137472016*	14	494621	*CYHR1-TONSL*	*rs41615143*	2	44981402	*RBM43(d)*	−0.008	60.69	23
*rs136580003*	14	399818	*ARHGAP39*	*rs137059769*	17	63245988	*C12orf76* (u)	0.005	60.69	24
*rs109146371*	14	465742	*PPP1R16A*	*rs43304498*	2	41451308	*KCNJ3-GALNT13*	−0.009	59.97	25
*rs110984572*	14	468124	*PPP1R16A-FOXH1*	*rs41615143*	2	44981402	*RBM43*(d)	−0.008	59.97	26
*rs137727465*	14	487527	*CYHR1*	*rs41615143*	2	44981402	*RBM43*(d)	0.008	59.97	27
*rs137727465*	14	487527	*CYHR1*	*rs109961025*	2	54049623	*KYNU-5S-rRNA*	0.010	59.97	28

Pos is the chromosome position of the SNP.

**Table 4 ijms-25-00674-t004:** Top 20 inter-chromosome A×A effects of the Chr14b1 region.

SNP-1	Chr-1	Pos-1	Gene-1	SNP-2	Chr-2	Pos-2	Gene-2	Effect	log10(1/p)	Rank
*rs3423093141*	14	2282659	Chr14b1	*rs42368654*	3	3924620	*LMX1A* (d)	0.0191	47.69	264
*rs3423094258*	14	2330431	Chr14b1	*rs42368654*	3	3924620	*LMX1A* (d)	−0.0182	55.03	81
*rs3423094258*	14	2330431	Chr14b1	*rs41981850*	21	41462728	*SCFD1-COCH*	−0.0126	45.14	359
*rs3423094258*	14	2330431	Chr14b1	*rs109787816*	23	30347149	*ZSCAN12*	−0.0272	50.30	182
*rs3423357679*	14	2350879	Chr14b1	*rs42368654*	3	3924620	*LMX1A* (d)	0.0178	57.12	50
*rs3423357679*	14	2350879	Chr14b1	*rs41981850*	21	41462728	*SCFD1-COCH*	0.0132	55.03	82
*rs3423357679*	14	2350879	Chr14b1	*rs109787816*	23	30347149	*ZSCAN12*	0.0248	45.14	360
*rs3423357679*	14	2350879	Chr14b1	*rs135435373*	31	6923026	*bta-mir-2285bj-1* (u)	0.0080	48.33	241
*rs136475864*	14	2372575	Chr14b1	*rs42368654*	3	3924620	*LMX1A* (d)	0.0161	52.97	126
*rs134537992*	14	2421119	Chr14b1	*rs43319812*	2	112888145	*DOCK10*	−0.0136	45.14	361
*rs134537992*	14	2421119	Chr14b1	*rs42368654*	3	3924620	*LMX1A* (d)	0.0187	64.37	5
*rs134537992*	14	2421119	Chr14b1	*rs109376678*	8	56181140	*TLE4(d)*	−0.0137	45.14	362
*rs134537992*	14	2421119	Chr14b1	*rs109489404*	8	56297348	*TLE4(d)*	−0.0138	45.77	328
*rs134537992*	14	2421119	Chr14b1	*rs133536911*	20	30612345	*FGF10 (d)*	0.0144	50.30	183
*rs134537992*	14	2421119	Chr14b1	*rs41940594*	20	35354207	*FYB1-RICTOR*	0.0140	48.33	242
*rs134537992*	14	2421119	Chr14b1	*rs41981850*	21	41462728	*SCFD1-COCH*	0.0136	59.25	32
*rs134537992*	14	2421119	Chr14b1	*rs109787816*	23	30347149	*ZSCAN12*	0.0250	46.41	304
*rs134537992*	14	2421119	Chr14b1	*rs109277263*	29	41499833	*LOC522784*	−0.0135	45.14	363
*rs134537992*	14	2421119	Chr14b1	*rs135435373*	31	6923026	*bta-mir-2285bj-1* (u)	0.0077	45.14	364
*rs41661929*	14	6113669	Chr14b2	*rs136387741*	31	87757884	*CLCN5*	−0.0049	45.77	330

Pos is the chromosome position of the SNP. Chr31 is the nonrecombining region of ChrX.

**Table 5 ijms-25-00674-t005:** Top 20 inter-chromosome A×A effects of the Chr14b2 region.

SNP-1	Chr-1	Pos-1	Gene-1	SNP-2	Chr-2	Pos-2	Gene-2	Effect	log10(1/p)	Rank
*rs132788949*	14	2867641	*PTK2*	(no rs number)	31	25156387	blank	−0.0053	46.41	305
*rs41624797*	14	2929132	*PTK2*	(no rs number)	31	25156387	blank	0.0054	48.98	214
*rs41624797*	14	2929132	*PTK2*	*rs135542379*	31	24950173	blank	−0.0051	47.04	283
*rs41624797*	14	2929132	*PTK2*	*rs41626477*	31	9512588	*TENM1*	0.0048	45.14	365
*rs41624797*	14	2929132	*PTK2*	*rs110945141*	5	36089282	*TMEM117*	0.0062	44.52	394
*rs41624797*	14	2929132	*PTK2*	*rs110881559*	2	133918945	*TAS1R2-PAX7*	0.0047	43.90	446
*rs55617160*	14	3439565	*TRAPPC9*	(no rs number)	31	25156387	blank	0.0052	46.41	306
*rs55617160*	14	3439565	*TRAPPC9*	*rs135542379*	31	24950173	blank	−0.0050	45.14	366
*rs55617160*	14	3439565	*TRAPPC9*	*rs110945141*	5	36089282	*TMEM117*	0.0062	43.90	447
*rs55617160*	14	3439565	*TRAPPC9*	*rs41626477*	31	9512588	*TENM1*	0.0048	43.90	448
*rs135838690*	14	3687442	*KCNK9*	*rs42368654*	3	3924620	*LMX1A* (d)	0.0151	45.77	329
*rs110822835*	14	3710917	*KCNK9*	*rs110945141*	5	36089282	*TMEM117*	0.0062	44.52	395
*rs110143087*	14	3738219	*KCNK9* (d)	*rs110945141*	5	36089282	*TMEM117*	0.0065	49.64	192
*rs110143087*	14	3738219	*KCNK9* (d)	*rs133552324*	10	35535274	*GPR176*	−0.0071	46.41	307
*rs110281272*	14	4021974	*KCNK9* (d)	*rs42477574*	5	34302710	*SCAF11*	−0.0056	45.14	367
*rs110281272*	14	4021974	*KCNK9* (d)	*rs136387741*	31	87757884	*CLCN5*	0.0050	45.14	368
*rs110281272*	14	4021974	*KCNK9* (d)	*rs136157041*	31	87819894	*CLCN5* (d)	0.0049	45.14	369
*rs110281272*	14	4021974	*KCNK9* (d)	*rs42477555*	5	34282642	*SCAF11*	0.0056	44.52	396
*rs110979942*	14	4543775	*FAM135B*	*rs109127443*	16	16191164	*5S-rRNA* (d)	−0.0100	44.52	397
*rs42306021*	14	4858211	*FAM135B* (d)	*rs135542883*	31	114523882	blank	0.0073	44.52	398

Pos is the chromosome position of the SNP. Chr31 is the nonrecombining region of ChrX.

**Table 6 ijms-25-00674-t006:** Top inter-chromosome A×A effects of the Chr05 and Chr20.

SNP-1	Chr-1	Pos-1	Gene-1	SNP-2	Chr-2	Pos-2	Gene-2	Effect	log10(1/p)	Rank
*rs110984572*	14	468124	*PPP1R16A-FOXH1*	*rs109208465*	5	5756462	*PHLDA1-BBS10*	0.0061	62.89	6
*rs137472016*	14	494621	*CYHR1-TONSL*	*rs109208465*	5	5756462	*PHLDA1-BBS10*	0.0061	62.89	7
*rs137727465*	14	487527	*CYHR1*	*rs109208465*	5	5756462	*PHLDA1-BBS10*	−0.0060	62.15	12
*rs109146371*	14	465742	*PPP1R16A*	*rs109208465*	5	5756462	*PHLDA1-BBS10*	−0.0059	59.25	31
*rs137727465*	14	487527	*CYHR1*	*rs137444512*	5	1183045	*LGR5*	0.0057	54.34	89
*rs137472016*	14	494621	*CYHR1-TONSL*	*rs137444512*	5	1183045	*LGR5*	−0.0057	54.34	90
*rs109146371*	14	465742	*PPP1R16A*	*rs137444512*	5	1183045	*LGR5*	0.0056	53.65	102
*rs110984572*	14	468124	*PPP1R16A-FOXH1*	*rs137444512*	5	1183045	*LGR5*	−0.0056	53.65	103
*rs110143087*	14	3738219	*KCNK9* (d)	*rs110945141*	5	36089282	*TMEM117*	0.0065	49.64	188
*rs110984572*	14	468124	*PPP1R16A-FOXH1*	*rs109706757*	5	4507251	*KCNC2*	−0.0054	48.33	219
*rs109208977*	14	243959	*ZNF250*	*rs136653182*	20	26615565	*ITGA1* (d)	0.0059	57.12	50
*rs134537992*	14	2421119	blank	*rs133536911*	20	30612345	*FGF10* (d)	0.0144	50.30	181
*rs136939758*	14	146715	*OR10AG83* (u)	*rs136653182*	20	26615565	*ITGA1* (d)	0.0055	48.98	209
*rs109208977*	14	243959	*ZNF250*	*rs133862450*	20	26701720	*ITGA1* (d)	−0.0050	48.98	210
*rs109208977*	14	243959	*ZNF250*	*rs135333478*	20	27147364	blank	0.0052	48.98	211
*rs109208977*	14	243959	*ZNF250*	*rs136075841*	20	28123462	*U6-PARP8*	−0.0052	48.98	212
*rs109968515*	14	490055	*CYHR1*	*rs135236809*	20	23802929	*MTREX*	0.0051	48.33	238
*rs136939758*	14	146715	*U6-OR10AG83*	*rs29024419*	20	28231492	*U6-PARP8*	−0.0054	48.33	239
*rs134537992*	14	2421119	blank	*rs41940594*	20	35354207	*FYB1-RICTOR*	0.0140	48.33	240
*rs136939758*	14	146715	*OR10AG83* (u)	*rs132937608*	20	28111718	*PARP8* (u)	−0.0055	47.69	258

Pos is the chromosome position of the SNP.

**Table 7 ijms-25-00674-t007:** Patterns of A×A epistasis effects between Chr14a and the SGN region of Chr06.

SNP-1	Gene-1	SNP-2	Gene-2	AC1	aa1	AC2	aa2	AC3	aa3	AC4	aa4
*rs109146371*	*PPP1R16A*	*rs42766480*	*GC-NPFFR2*	1_1	0.0045	2_1	0.0006	2_2	−0.0007	1_2	−0.0030
*rs109146371*	*PPP1R16A*	*rs110352004*	*GC-NPFFR2*	1_2	0.0034	2_2	0.0003	2_1	−0.0005	1_1	−0.0031
*rs110984572*	*PPP1R16A-FOXH1*	*rs42766480*	*GC-NPFFR2*	2_1	0.0046	1_1	0.0006	1_2	−0.0007	2_2	−0.0031
*rs110984572*	*PPP1R16A-FOXH1*	*rs110352004*	*GC-NPFFR2*	2_2	0.0035	1_2	0.0003	1_1	−0.0005	2_1	−0.0032
*rs110984572*	*PPP1R16A-FOXH1*	*rs109901151*	*SLC4A4*	2_2	0.0034	1_2	0.0003	1_1	−0.0005	2_1	−0.0031
*rs137727465*	*CYHR1*	*rs42766480*	*GC-NPFFR2*	1_1	0.0046	2_1	0.0006	2_2	−0.0007	1_2	−0.0031
*rs137727465*	*CYHR1*	*rs110352004*	*GC-NPFFR2*	1_2	0.0035	2_2	0.0003	2_1	−0.0005	1_1	−0.0032
*rs137727465*	*CYHR1*	*rs109901151*	*SLC4A4*	1_2	0.0034	2_2	0.0003	2_1	−0.0005	1_1	−0.0031
*rs137472016*	*CYHR1-TONSL*	*rs42766480*	*GC-NPFFR2*	2_1	0.0046	1_1	0.0006	1_2	−0.0007	2_2	−0.0031
*rs137472016*	*CYHR1-TONSL*	*rs110352004*	*GC-NPFFR2*	2_2	0.0035	1_2	0.0003	1_1	−0.0005	2_1	−0.0032
*rs137472016*	*CYHR1-TONSL*	*rs109901151*	*SLC4A4*	2_2	0.0034	1_2	0.0003	1_1	−0.0005	2_1	−0.0031
*rs211309638*	*ADCK5-SLC52A2*	*rs110352004*	*GC-NPFFR2*	2_2	0.0036	1_2	0.0004	1_1	−0.0006	2_1	−0.0034
*rs109421300*	*DGAT1*	*rs109901151*	*SLC4A4*	1_2	0.0036	2_2	0.0005	2_1	−0.0008	1_1	−0.0032
*rs109421300*	*DGAT1*	*rs109512265*	*SLC4A4*	1_2	0.0036	2_2	0.0005	2_1	−0.0008	1_1	−0.0033
*rs109421300*	*DGAT1*	*rs110953922*	*SLC4A4*	1_2	0.0036	2_2	0.0005	2_1	−0.0008	1_1	−0.0033
*rs109421300*	*DGAT1*	*rs110352004*	*GC-NPFFR2*	1_2	0.0036	2_2	0.0006	2_1	−0.0009	1_1	−0.0033
*rs109421300*	*DGAT1*	*rs137302420*	*SLC4A4*	1_1	0.0036	2_1	0.0005	2_2	−0.0008	1_2	−0.0033
*rs109421300*	*DGAT1*	*rs110434046*	*GC-NPFFR2*	1_2	0.0032	2_2	0.0006	2_1	−0.0015	1_1	−0.0039
*rs109421300*	*DGAT1*	*rs137844449*	*NPFFR2*	1_2	0.0038	2_2	0.0004	2_1	−0.0003	1_1	−0.0018
*rs109421300*	*DGAT1*	*rs109034709*	*NPFFR2*	1_2	0.0032	2_2	0.0006	2_1	−0.0015	1_1	−0.0038

AC1–AC4 are the four allelic combinations of the two loci, and aa1–aa4 are the A×A epistasis values of AC1–AC4.

**Table 8 ijms-25-00674-t008:** Patterns of A×A epistasis effects of Chr14b.

SNP-1	Gene-1	SNP-2	Gene-2	AC1	aa1	AC2	aa2	AC3	aa3	AC4	aa4
*rs134537992*	Chr14b1	*rs42368654*	*LMX1A* (d)	1_1	0.0124	2_2	0.0001	1_2	−0.0007	2_1	−0.0055
*rs3423357679*	Chr14b1	*rs42368654*	*LMX1A* (d)	1_1	0.0120	2_2	0.0001	1_2	−0.0007	2_1	−0.0050
*rs3423094258*	Chr14b1	*rs42368654*	*LMX1A* (d)	2_1	0.0114	1_2	0.0001	2_2	−0.0008	1_1	−0.0059
*rs136475864*	Chr14b1	*rs42368654*	*LMX1A* (d)	1_1	0.0116	2_2	0.0000	1_2	−0.0006	2_1	−0.0038
*rs3423093141*	Chr14b1	*rs42368654*	*LMX1A* (d)	1_1	0.0150	2_2	0.0001	1_2	−0.0011	2_1	−0.0030
*rs135838690*	*KCNK9*	*rs42368654*	*LMX1A* (d)	1_1	0.0103	2_2	0.0001	1_2	−0.0005	2_1	−0.0043
*rs134537992*	Chr14b1	*rs41981850*	*SCFD1-COCH*	1_1	0.0094	2_2	0.0001	1_2	−0.0009	2_1	−0.0032
*rs3423357679*	Chr14b1	*rs41981850*	*SCFD1-COCH*	1_1	0.0094	2_2	0.0001	1_2	−0.0009	2_1	−0.0028
*rs134537992*	Chr14b1	*rs133536911*	*FGF10_U6*	1_1	0.0100	2_2	0.0001	1_2	−0.0007	2_1	−0.0037
*rs3423094258*	Chr14b1	*rs109787816*	*ZSCAN12*	2_1	0.0172	1_2	0.0001	2_2	−0.0006	1_1	−0.0094
*rs134537992*	Chr14b1	*rs109787816*	*ZSCAN12*	1_1	0.0162	2_2	0.0001	1_2	−0.0005	2_1	−0.0083
*rs110143087*	*KCNK9* (d)	*rs110945141*	*TMEM117*	2_2	0.0037	1_1	0.0004	2_1	−0.0007	1_2	−0.0018
*rs134537992*	Chr14b1	*rs41940594*	*FYB1_RICTOR*	1_1	0.0099	2_2	0.0001	1_2	−0.0007	2_1	−0.0033
*rs3423357679*	Chr14b1	*rs135435373*	*bta-mir-2285bj-1* (u)	1_1	0.0051	2_2	0.0002	2_1	−0.0010	1_2	−0.0017
*rs41624797*	*PTK2*	*rs135542379*	blank	2_1	0.0016	1_2	0.0007	1_1	−0.0003	2_2	−0.0025
*rs110143087*	*KCNK9* (d)	*rs133552324*	*GPR176*	2_1	0.0048	1_2	0.0002	2_2	−0.0006	1_1	−0.0016
*rs134537992*	Chr14b1	*rs109489404*	blank	1_2	0.0094	2_1	0.0001	1_1	−0.0008	2_2	−0.0035
*rs134539615*	*ZFAT* (d)	*rs29016827*	*STXBP6*	1_2	0.0016	2_1	0.0008	1_1	−0.0009	2_2	−0.0015
*rs134539615*	*ZFAT* (d)	*rs109853041*	*STXBP6*	1_2	0.0016	2_1	0.0009	1_1	−0.0009	2_2	−0.0015
*rs41661929*	blank	*rs136387741*	*CLCN5*	1_2	0.0012	2_1	0.0012	2_2	−0.0008	1_1	−0.0018

AC1–AC4 are the four allelic combinations of the two loci, and aa1–aa4 are the A×A epistasis values of AC1–AC4.

## Data Availability

The original genotype data are owned by third parties and maintained by the Council on Dairy Cattle Breeding (CDCB). A request to the CDCB is necessary to obtain data access for research, which may be sent to João Dürr, CDCB Chief Executive Officer (joao.durr@cdcb.us). All other relevant data are available in the manuscript and Appendix A.

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
