# Peer review of "A Million-Cow Validation of a Chromosome 14 Region Interacting with All Chromosomes for Fat Percentage in U.S. Holstein Cows"

_ijms, 2024, doi:10.3390/ijms25010674_

Round 1
Reviewer 1 Report
Comments and Suggestions for Authors
The authors have presented a compelling story from a study of unprecedented scale and complexity at least in the animal science realm. Most of my comments on the attached pdf file are minor suggestions intended to improve readability or clarify small concerns. I would suggest the authors take a little bit of time in the Introduction to include WHY the AxA interactions you have detected are important for the genetic evaluations in dairy cattle. I think the authors have an opportunity to line out how future studies are needed like this to make confident conclusions on these complex interactions. Make a case to the average reader why they should care. In general, I applaud the authors efforts to bring this kind of evaluation to light!

Author Response
We thank you for your positive review.
The authors have presented a compelling story from a study of unprecedented scale and complexity at least in the animal science realm. Most of my comments on the attached pdf file are minor suggestions intended to improve readability or clarify small concerns. I would suggest the authors take a little bit of time in the Introduction to include WHY the AxA interactions you have detected are important for the genetic evaluations in dairy cattle. I think the authors have an opportunity to line out how future studies are needed like this to make confident conclusions on these complex interactions. Make a case to the average reader why they should care. In general, I applaud the authors efforts to bring this kind of evaluation to light!
We appreciate these comments. Just to clarify, the introduction was a short review of what was found: AxA effects from the previous study, not to promote such effects.
Clearly you have the most significant data set to evaluate completely the impact and gene action of the DGAT1 region. Since the broader audience that may read this does not necessarily understand, could you add some general comments about why AxA interactions are important and better yet how they should be used to improve the genetic evaluations?
The article made the point about the power of the large sample size at the end of the introduction and in the abstract and conclusions. We intentionally did not mention genomic prediction because that is a large and separate topic.
Line 34-38, Clarify that in this sentence this is the additive effect you are describing is for the Chr14 with DGAT1 region? Or is this the scale of the overall whole genome additive effects for these traits?
The sentence should be clear because it says additive effects from a previous GWAS, so that the effects were genome-wide effects. The point here was how much more significant FPC effects were than for other traits, not identifying the locations of specific effects.
Lines 50-53, This is a long compound sentence. Consider breaking into two?
The sentence is shortened.
Line 166, ‘of Chr14a’ is added.
Line 248, ‘,’ is changed to ‘.’. Thanks.
Line 293, ‘eighter’ is deleted. Thanks.
Reviewer 2 Report
Comments and Suggestions for Authors I am reviewing the manuscript entitled 'A million-cow validation of a chromosome 14 region interacting with all chromosomes for fat percentage in U.S. Holstein cows' for consideration to be published in International Journal of Molecular Sciences. The manuscript investigates a Chr14 region with a size of about 9.38 Mb which has a significant inter‐chromosome additive × additive (A×A) effects with all chromosomes for fat percentage in U.S. Holstein cows. The authors used GWAS to analysis two sub-regions of the significant A×A effects of Chr14. Results support FPC was affected by genome‐wide allele ×allele interactions centered in the 9.38 Mb Chr14 region. There are some major points to be improved on this manuscript: 1. The author do not separate the discussion from the results. 2. Line16-22 must be improved. 3. “Pos” and “Rank” in table are not annotated. 4. Line 264 is not appropriate.Author Response
We thank the reviewer for a helpful review.
- The author do not separate the discussion from the results.
The combination of results with discussion is an appropriate format for this article due to the large number of details. Most of the details were immediately followed by discussions. Separating the results from their immediately discussion would make the connections between the results and their discussions less tractable.
- Line16-22 must be improved.
These lines are now re-written as:
The PPP1R16A-FOXH1-CYHR1-TONSL (PFCT) region of Chr14a (29 Kb in size) with four SNPs had the largest number of inter-chromosome A×A effects (1141 pairs) with all chromosomes including the most significant inter-chromosome A×A effects. The SLC4A4-GC-NPFFR2 (SGN) region of Chr06 known to have highly significant additive effects for some production, fertility and health traits specifically interacted with the PFCT region and a Chr14a region with CPSF1, ADCK5, SLC52A2, DGAT1, SMPD5 and PARP10 (CASDSP) known to have highly significant additive effects for milk production traits. The most significant effects were between a SNP in SGN and four SNPs in PFCT.
- “Pos” and “Rank” in table are not annotated.
A footnote for ‘Pos’ is added. Rank is self-explanatory and should not need to be defined (none of our previous GWAS publications provided a definition for rank).
- Line 264 is not appropriate.
This line is modified as: Inter-chromosome A×A effects of Chr20 and Chr05 interacting with Chr14.